# Factors Associated with Depression and Anxiety in Adults ≥60 Years Old during the COVID-19 Pandemic: A Systematic Review

**DOI:** 10.3390/ijerph182211859

**Published:** 2021-11-12

**Authors:** Gianluca Ciuffreda, Sara Cabanillas-Barea, Andoni Carrasco-Uribarren, María Isabel Albarova-Corral, María Irache Argüello-Espinosa, Yolanda Marcén-Román

**Affiliations:** 1Faculty of Health Sciencies, University of Zaragoza, 50009 Zaragoza, Spain; isabelalbarova@gmail.com; 2Élite Fisioterapia, 50018 Zaragoza, Spain; scabanillas@uic.es (S.C.-B.); acarrasco@uic.es (A.C.-U.); 3Faculty of Medicine and Health Sciences, Universitat Internacional de Catalunya, 08195 Sant Cugat del Vallès, Spain; 4Internal Medicine, Hospital Clínico Universitario “Lozano Blesa”, 50009 Zaragoza, Spain; irache749@gmail.com; 5Faculty of Medicine, Institute of Research of Aragon, University of Zaragoza, 50009 Zaragoza, Spain

**Keywords:** COVID-19, depression, anxiety, mental health, older adults, aged, associated factors, risk factors

## Abstract

COVID-19 represents a threat to public health and the mental health of the aged population. Prevalence and risk factors of depression and anxiety have been reported in previous reviews in other populations; however, a systematic review on the factors associated with depression and anxiety in older adults is not currently present in the literature. We searched PubMed, Embase, Scopus, ProQuest Psychology Database, Science Direct, Cochrane Library and SciELO databases (23 February 2021). The results were obtained by entering a combination of MeSH or Emtree terms with keywords related to COVID-19, elderly, depression and anxiety in the databases. A total of 11 studies were included in the systematic review. Female gender, loneliness, poor sleep quality and poor motor function were identified as factors associated with both depression and anxiety. Aspects related to having a stable and high monthly income represent protective factors for both depression and anxiety, and exercising was described as protective for depression. This study synthesised information and analysed the main factors associated with depression and anxiety in the older population during the COVID-19 pandemic. However, the cross-sectional design of most of the included studies does not allow a causal relationship between the factors analysed and depression or anxiety.

## 1. Introduction

The beginning of the COVID-19 pandemic marked a radical change in people’s lives, becoming a real threat to public health worldwide.

Many of the preventive measures adopted in different countries had among their main priorities protecting older people due to the severity of the manifestation of the disease and the high morbidity and mortality rates in this population [1,2,3]. This situation can be expressed in the mortality rate increase among the population over 60 years of age, which was described to be up to 5 times higher than in other age groups [4].

Inevitably the pandemic and its containment measures had a relevant impact on the sociocultural, economic, and psychological spheres [5,6,7,8,9]. The prevention and mitigation of the disease itself have been the main focus of attention. Despite this, the minimisation of the negative consequences of isolation at the psychosocial level has often been in the background. In many countries, the public health policies adopted different measures, such as home lockdowns and closures of social gathering places [10]. In older people, social isolation is especially relevant due to the possible decrease in different functional aspects of the person and an increase in the difficulty in carrying out daily activities [11]. The consequences on mental health have been widely studied by many authors, who have described how the context of COVID-19 can affect the geriatric population’s mental state with disorders such as depression, anxiety, stress or insomnia [5,6,12,13,14,15,16]. According to previous research, the prevalence of depression in older adults showed a significant increase from 7.2% to 19.8% since the beginning of the pandemic [17] and psychogeriatric admissions increased by more than 21% [18]. The factors associated with different psychological conditions during the pandemic were investigated in a meta-analysis by Wang et al. [6], which focused predominantly on the general population and reported that female gender, being aged <35, lower socio-economic status, higher risk of COVID-19 infection and longer media exposure were associated with psychological distress. The risk and protective factors of depression in older adults were described in a recent systematic review, which more homogeneously identified impairment, sleep disorders and chronic diseases as main risk factors, and physical activity as a protective factor; however, this review was not focused on the period of the COVID-19 pandemic, as the included studies were published between January 2000 and March 2020 [19]. To the best of our knowledge, no systematic reviews have been published to analyse the factors associated with depression and anxiety in the aged population during COVID-19. The identification and understanding of these factors represent a relevant aspect of minimising the impact on the psychosocial sphere of the adverse effects that the preventive and containment measures adopted during the pandemic may produce.

Hence, this study aimed to conduct a systematic review to explore the possible factors associated with depression and anxiety during the COVID-19 pandemic in a population older than 60 years.

We hypothesised that female gender, older age, poorer health conditions, loneliness, and a lower socio-economic status may be associated with depression and anxiety during the pandemic.

## 2. Materials and Methods

The presented systematic review was conducted according to the Preferred Reporting Items for Systematic Reviews and Meta-Analyses [20]. It was registered in the database for PROSPERO systematic reviews with the number CRD42021267998.

### 2.1. Search Strategy

The bibliographic search process was carried out in February 2021 in the following databases: PubMed, Embase, Scopus, ProQuest Psychology Database, Science Direct, Cochrane Library and SciELO. The results were obtained by entering a combination of MeSH or Emtree terms with keywords related to COVID-19, older adults, depression and anxiety in the databases. The search terms were combined with the Boolean operators AND and OR. To optimise the results, adjustments were made in the search procedure depending on the database used. The employed search terms and the search strategy used for each database are specified in the Appendix A. 

### 2.2. Eligibility Criteria

This systematic review included studies that analysed factors associated with depression or anxiety during the period of the COVID-19 pandemic, indicating that the subjects were ≥60 years of age, with original data, studies with a longitudinal or cross-sectional design, published in English, providing access to the full text, and expressing the predictive value of the factors associated with depression and/or anxiety with Odds Ratios (ORs) and their respective 95% confidence interval (CI).

The articles excluded for this review were those that did not show data on factors associated with depression and/or anxiety during the period of the COVID-19 pandemic, those that did not show original data (reviews, meta-analysis, opinion articles, study protocols, etc.) and clinical cases or case series. Studies with analysis stratified by age were excluded if the age of the total sample was <60.

### 2.3. Study Selection

The sequence for selecting the studies was, first, a search by combining MeSH terms and keywords in the different databases. Subsequently, duplicate articles were eliminated, and once the title and abstract had been read, potentially relevant articles were identified. When reading the entire document, the inclusion and exclusion criteria proposed for this review were considered.

Two independent reviewers oversaw the search process, selecting articles, and data extraction (G.C. and S.C.-B.). A third investigator (A.C.-U.) resolved any doubts of or discrepancies between the two principal investigators during this process.

For each article in this review, the following data were extracted: author, year of publication, type of study, characteristics of the participants, number of subjects, outcome measures and results obtained (Table 1 and Table 2).

When possible, the ORs obtained with multivariate analysis were reported; if this analysis was not carried out, the ORs were reported from other analyses performed in the study. The statistical analysis of each study is specified in Table 1 and Table 2.

### 2.4. Evaluation of the Quality of the Studies

The quality assessment of the included articles was carried out independently by the researchers G.C. and S.C.-B., with the researcher A.C.-U. in charge of resolving any discrepancies. The assessment was performed with the Joanna Briggs Institute scales for non-randomised studies [32]. These tools were previously used to determine the quality of cross-sectional and cohort studies included in other systematic reviews on factors associated with psychological stress resulting from the coronavirus disease [6] and other emerging viruses [33]. The domains of the tools focus on the characteristics of the study population, the comparability of the groups, the validity and reliability of the exposure or outcome measurement method. The scales for cross-sectional and cohort studies evaluate 8 and 11 items, respectively (Table 3 and Table 4).

## 3. Results

### 3.1. Study Selection 

Our search (February 2021) identified 3756 citations from six databases. After removing duplicates, 2689 studies remained. From the 2689 studies, 2415 papers were excluded after screening the title and abstract. After reading 274 full-text papers, 263 were eliminated for not meeting the inclusion criteria. Finally, 11 studies were included in the current systematic review (Figure 1). The methodological quality of the included studies is summarised in Table 3 and Table 4.

### 3.2. Study Characteristics 

The detailed characteristics of the eleven included papers are listed in Table 1 and Table 2.

Ten papers were cross-sectional studies [21,22,23,24,25,26,27,29,30,31], and one was a retrospective longitudinal study [28].

Ten of the included studies investigated factors associated with depression [21,22,23,24,25,26,27,28,29,30], six with anxiety [21,23,24,26,27,30], and one unified both psychological outcomes in a single variable expressed as “depression or anxiety” [31]. Additionally, one of the studies analysed the associations with comorbid depression and anxiety [26].

Among the included studies, five were from Europe [21,22,23,24,31], four were from Asia [25,26,27,30], one was from North America [28], and the last was from Latin America [29]. Specifically, the countries where the patients were recruited were: Italy [22,24], Spain [21], United Kingdom [23], France [31], Japan [27], Turkey [30], Vietnam [25], China [26], Canada [28] and various Spanish speaking countries of Latin America [29].

The majority of studies were conducted among the general population [21,23,28,30,31]. However, some of the papers included participants with specific characteristics: adults with mild cognitive impairment [22,24], psychiatric disorders [26], Parkinson’s disease [27], cardiometabolic disorders [29] and COVID-19 [25].

A variety of psychometric instruments were used to measure symptoms of anxiety and depression. The most often used tool was the Patient Health Questionnaire for depression (PHQ-9), used in four studies [25,26,27,31], followed by the Geriatric Depression Scale (GDS), used in two studies in the 5-item version [22,24] and one study in the 15-item version [30]. The following tools were used in the remainder of the studies: the Depression Rating Scale (DRS) [28], the Depression items of the Hospital Anxiety and Depression Scale (HADS) [23], the diagnostic criteria of the Diagnostic and Statistical Manual for Mental Disorders (DSM-5) [29] and the Depression items of the Depression Anxiety Stress Scale (DASS) [21]. To assess anxiety, four studies used the Generalized Anxiety Disorder (GAD-7) [24,26,27,31], one the Geriatric Anxiety Inventory (GAI) [30], one the anxiety items of the HADS [23] and the last used the anxiety items of the DASS [21].

All OR values reported in Table 1 and Table 2 were obtained from the multivariate analysis [21,22,23,24,25,26,27,28,29,30,31].

### 3.3. Gender

Gender represents the socio-demographic factor most frequently associated with depression and anxiety. Its relationship with depression was analysed in seven studies [21,23,27,28,29,30,31] and with anxiety in five studies [21,23,27,30,31].

Five papers screened showed that the female gender is associated significantly with depression or anxiety, with Odds Ratio (OR) values ranging from 1.72 to 2.11 for depression [21,28,29,31] and from 1.98 to 3.32 for anxiety [21,30,31].

Robb et al. [23] investigated the association between female gender and changes in depression and anxiety symptoms; they found an association between female gender and worsening depression and anxiety since the pandemic.

Only one study included in this review described male gender as a significant risk factor for anxiety with OR: 17.12 (1.13–257.27) in Parkinson’s disease patients; in the same study, male gender was also reported as a risk factor for depression with OR: 5.66 (0.51–62.47). However, the association was not statistically significant in the last case [27].

### 3.4. Age

A total of five publications analysed associations between age and depression or anxiety [21,23,27,28,30]. Two studies reported a significant association between age and lower odds for anxiety or depression [21,23]. Specifically, Robb et al. [23] found that with every five-year increase in age, there was a 19% (OR: 0.81 [0.77–0.85]) and 22% (OR: 0.78 [0.75–0.83]) lower risk of reporting feeling worse with regard to components of depression and anxiety, respectively; and Bobes-Bascarán et al. [21] showed that younger age was a protective factor for anxiety (OR: 0.876 [0.800–0.960]). In addition, Kitani-Morii et al. [27] reported that being aged between 70 and 79, in comparison to having an age of <70 years, represented a factor associated with depression in Parkinson’s disease patients (OR: 0.61 [0.05–0.08]). The remainder of the associations were not statistically significant.

### 3.5. Physical and Mental Conditions

Three studies analysed the association between different physical pathologies and depressive symptoms in older adults [22,26,30]. Among these, only Carlos et al. [22] found a statistically significant association between general health disorders and depression (OR: 2.45 [1.16–5.16]). Anxiety had a significant association with suffering flu symptoms [22], having severe physical diseases [26] and having COVID-19 symptoms for more than 14 days [21] (OR: 4.01 [1.13–14.24]; 1.57 [1.05–2.35]; 7.584 [1.398–41.146], respectively). 

Presenting past or current psychiatric disorders [21] and a history of anxiety [31] were predictive factors for depression. In contrast to these data, two studies included in this review suggest that having dementia [28] or other psychiatric disorders [26] may represent a protective factor for depression (OR: 0.69 [0.48–0.99]; 0.50 [0.35–0.71]). 

Two studies reported a significant association between anxiety and these two factors: the presence of subjective cognitive decline [24] and mental disorders [21] (OR: 6.202 [3.005–12.799]; 4.39 [1.03–18.69], respectively).

One study reported that having schizophrenia and other psychiatric disorders may represent a protective factor for the combination of depression and anxiety (OR: 0.50 [0.26–0.97]; 0.53 [0.38–0.73]) [26].

### 3.6. Sleep Quality

Different studies investigated the association of factors related to sleep quality with depression and anxiety. Carlos et al. [22] reported that the presence of sleep disorders is significantly associated with depression (OR: 2.29 [1.06–4.93]). In line with these results, Li et al. [26] showed that insomnia is associated with depression, anxiety and the combination of both of them (OR: 1.29 [1.24–1.34]; 1.15 [1.12–1.18]; 1.19 [1.16–1.23]). One study revealed that poor sleep quality has also been associated with worsening depressive symptoms and anxiety since the pandemic, with higher odds values for poorer sleep quality [23].

### 3.7. Loneliness, Social Isolation and Personal Relationships

Several publications analysed the relationship between depression and anxiety with different factors within loneliness and social isolation.

Two studies reported that living alone may significantly predict depression and increase depressive symptoms [23,24]. In addition to living alone, Robb et al. [23] analysed the feeling of loneliness, finding that feeling lonely more often has a strong association with experiencing a worsening of depressive symptoms (OR: 17.24 [13.20–22.50]) and anxiety (OR: 10.85 [8.39–14.03]).

The impact of personal relationships in older people was explored in different studies. Robb et al. [23] suggest that being single, widowed or divorced represents a factor significantly associated with worsening depression after the beginning of the pandemic (OR: 1.37 [1.17–1.59]); by comparison, Bobes-Bascarán et al. [21] report that never having been married represented a protective factor for depression (OR: 0.665 [0.456–0.970]). Bérard et al. [31] show in their study that the worsening of the relationship with the partner from the beginning of the confinement is significantly associated with depression or anxiety (OR: 5.24 [2.11–13.0]). 

Social media contact with family and friends is associated with improving and worsening anxiety symptoms [23]. Having friends or family with COVID-19 represents a factor significantly associated with depression (OR: 1.631 [1.247–2.132]) [21].

### 3.8. Other Factors

Regarding socio-economic factors, a study carried out in Spain reports that being a civil servant and being retired represent protective factors for the presence of depression at the time of the pandemic (OR: 0.530 [0.293–0.957]; OR: 0.539 [0.311–0.934]) [21]; another study, in Turkey, assessed the amount of monthly income, and found that having a higher income was a significant protective factor for both depression and anxiety (OR: 0.13 [0.04–0.44]; OR: 0.07 [0.02–0.35]) [30].

Several authors evaluated aspects of motor functions. In subjects with Parkinson’s disease, the second part of the MDS-UPDRS questionnaire shows a significant association with depression and anxiety (OR: 1.31 [1.04–1.66]; 1.36 [1.07–1.72]) [27]. Other authors reported that poorer functional mobility is associated with depression (OR: 1.86 [1.35–2.56]) [28].

Physical activity is another variable that has been studied. Two studies found data in the same line. One author describes that less physical activity is associated with depression [29], whereas another reports that exercising regularly may represent a protective factor (OR: 0.30 [0.12–0.72]) [22].

Do et al. [25] analysed the association between depression and health literacy in people with and without COVID-19, finding a significant association in patients with COVID-19 (OR: 0.91 [0.87–0.94]).

Two publications analysed nutritional factors: Piskorz et al. [29] found significant associations of low consumption of fruit or vegetables and a reduced food intake with depression in patients with cardiometabolic disorders (OR: 1.46 [1.05–2.03]; 2.10 [1.68–2.62]); Bérard et al. [31] reported that a pre-confinement Diet Quality Score > the median of the sample presented OR values of 0.51 (0.31–0.85) for depression or anxiety.

## 4. Discussion

The COVID-19 pandemic had a considerable impact on people’s lives, and especially on those of older adults. Mental health has been an issue of public concern, with disorders such as depression or anxiety among the most relevant conditions [6,7]. To the best of our knowledge, the present study is the first systematic review investigating the factors associated with depression and anxiety during the COVID-19 pandemic in people aged ≥60. The data from ten cross-sectional studies and one longitudinal study were analysed.

The factors reported were predominantly related to socio-demographic characteristics or physical and mental health aspects. Among the socio-demographic factors, the female gender is the most frequently associated with depression and anxiety [21,23,27,28,29,30,31]. These findings are consistent with other reviews focused on the general population and healthcare workers during the pandemic [6,7] and more studies before COVID-19 [19,34,35]. Previous research suggests that this gender disparity may be mediated by variables that concern work, economic, educational, neuro-hormonal, psychological and genetic aspects [34,36,37,38]. The economic crisis caused by the pandemic, combined with a baseline situation of social inequity, may be a trigger for an increased adverse psychological response. Gender inequity has been a cause of suffering for many women worldwide, leading to worse mental health outcomes in different contexts [34,36], including the SARS epidemic in 2003 [39].

Regarding age, despite being the group of individuals most at risk of suffering from a severe form of the disease [1,4], and in contrast to our initial hypothesis, the results found in this review indicate that being older is a protective factor for depression or anxiety [21,23]. It is important to note that in our review, among the studies that analysed age as a predictor, three recruited their sample directly from older people [21,23,30], one selected long-term care home residents with a mean age of 81.4 (±11.5) years [28], and in one most of the participants were in the age range of 70–79 years (mean age 72.3 ± 10.9) [27]. Consequently, it must be considered that the comparisons between ages when calculating the OR were made within a relatively similar range. This may have led to obtaining less significant odds than comparing age groups with a greater difference. The results of a review carried out in the general population during the period of COVID-19 were in line with our data, and reported that younger ages presented a higher OR for both depression and anxiety [6].

By comparison, pre-pandemic studies analysing the relationship between age and mental health outcomes associate the presence of depression and anxiety with the advance in age, with a prevalence among the older population ranging between 3% and 15% for anxiety and up to 42% for depression [35,40]. A possible explanation for this difference may be that, in the context of COVID-19, multiple control measures involved social restrictions and reduction of working activity in many areas. Younger people may have been more affected by such a radical lifestyle change. They may experience greater fear regarding their occupational and economic future than a population already retired or close to the retirement age and whose daily routine may have been affected in a minor way. Supporting this, one of the studies included in this review identified being retired and aspects related to having stable and high incomes as protective factors for depression [21].

Physical and mental health have been previously associated with depression and anxiety [6,35,37]. In our review, the results regarding the factors related to physical health show some variability depending on the outcome. Among the studies that investigated the relationship between several physical health variables and depression, two did not find statistically significant associations [26,30]; only one study reported that suffering health problems may represent a risk factor for depressive symptoms [22]. By comparison, in our study, different physical conditions presented a greater number of significant associations with anxiety, suggesting that flu symptoms [24], severe physical illnesses [26] and having more than 14 days with COVID-19 symptoms [21] may represent possible risk factors. Mental health factors have several significant associations. However, whether these factors are associated with positive or negative outcomes may change according to the study: three studies identified the presence of current cognitive disorders as risk factors for depression and anxiety [21,24,31], whereas two others reported that the presence of different psychiatric diagnoses represented protective factors for depression [26,28] and comorbid depression and anxiety [26]. It is interesting to note that the data collected in most of these studies was self-reported through surveys, although it was not clearly stated how the presence/absence of pathology was defined, and data was not collected via a clinical examination by a professional; these factors may lead to differences in the results. The factors in the health sphere that were more consistently associated with both depression and anxiety among the studies were those related to the presence of low quality of sleep [22,23,26], and psychophysical comorbidity was among the most prevalent of the COVID-19 era [41].

Several studies in this review explored many aspects of loneliness, social isolation and personal relationships. According to our findings, living or feeling alone are associated with both depression and anxiety [23,24]. In line with these data, several of the included authors found that being single, widowed, or divorced, or experiencing a worsening of relationships with their partner, is associated with worse psychological outcomes [23,31]. Another study showed that never having been married represented a protective factor for the presence of depressive symptoms [21]. Social isolation during the pandemic was described in a previous review as a risk factor in other populations [7]. Indeed, people’s social sphere has been one of the most affected by COVID-19 [9]. Containment measures, together with public health policies, resulted in a drastic reduction in social life, which could nearly be eradicated in the case of a home lockdown. It may be expected that in a context where a large degree of the development of social relationships occurs within the household, living alone, without a partner or with bad relationships with partners may predispose individuals to the development of depressive symptoms or anxiety. However, these considerations about social relationships may not extend to contacts via technology. One study included in our review shows that levels of social media contacts did not significantly alter the risk of reporting worsening depression and that their relationship with anxiety was unclear [23].

Regarding the association between physical activity and depression, in our review, two studies consistently reported data associating a lower amount of physical activity with depression [29], and that exercising may represent a protective factor against it [22]. The association of physical activity with depression has previously been investigated in the older population. Our results are consistent with a review conducted prior to the pandemic, which reported that having low physical activity levels may represent a risk factor for depressive symptoms [42]. Additionally, it should be considered that increased levels of physical activity may lead to better motor function, especially in an aged population.

Previous research associated depression with aspects related to a lower motor function, such as less grip strength [43] or a slower gait speed [44]. The relationship of poorer function, disability and depression is often described as bidirectional, with the possibility that both factors influence each other [44]. In our study, the aspects of motor function that presented a significant association with depression were poorer functional mobility [28] and poorer motor capacity in subjects with Parkinson’s disease [27].

This review was subject to some limitations. First, only studies published in English were included. Considering the global nature of the COVID-19 pandemic, this may have led to the loss of information published in other languages. Including only peer-reviewed studies and not assessing the grey literature may have contributed to further loss of information. However, this criterion was chosen to establish a minimum quality standard for the publications to be included.

Furthermore, we must mention that our review only included studies in which the age of the total sample was ≥60 years, and excluded publications in which the sample was only presented in a stratified manner. This criterion was adopted considering that the population of interest was specified with terms related to older adults when conducting the bibliographic search. Moreover, it must be taken into account that the included studies were performed mostly within the first months since the beginning of the pandemic. Thus, considering the changes in public health policies and the evolution of the COVID-19 pandemic, the results of this systematic review can only be related to this period. Lastly, the design of the included studies must be considered. All the studies except one used a cross-sectional design, which does not allow information to be obtained to detect changes over time in the psychological state or estimate the real impact that the pandemic and its containment measures may have had on depression or anxiety. Similarly, it also limits the understanding of the causality between the factors and outcomes studied.

Our systematic review focused exclusively on describing the associated factors for depression and anxiety. However, the COVID-19 pandemic has led to multiple psychological conditions in the elderly population. Future studies may include other mental health outcomes, such as stress, insomnia or fear. Analysing the factors associated with a more significant number of disorders of the psychic sphere may positively impact the aspects that concern their prevention and treatment during the pandemic.

## 5. Conclusions

COVID-19 mainly affects older adults, as reflected in morbidity and mortality rates. Taking this into account, this review focused on the possible psychological aspect of the COVID-19 pandemic in this population by analysing the factors associated with depression and anxiety.

Female gender, loneliness, poor sleep quality and poor motor function were identified as factors associated with both depression and anxiety. Levels of physical activity or exercise were associated with depression, with lower levels of activity identified as risk factors and exercising regularly as a protective factor. Several physical health conditions may be associated with anxiety. Aspects related to having a stable and high monthly income represent protective factors for both depression and anxiety. A small number of studies suggested that being older may represent a protective factor for anxiety and depression; however, other studies did not find this association to be significant. Due to the lack of consistency between the studies analysed, the association between mental health status and depression or anxiety is unclear.

This study collected information and analysed the different factors associated with depression and anxiety in the aged population during the pandemic. However, the cross-sectional design of most of the included studies does not allow the formulation of a causal relationship between the factors analysed and depression or anxiety.

## Figures and Tables

**Figure 1 ijerph-18-11859-f001:**
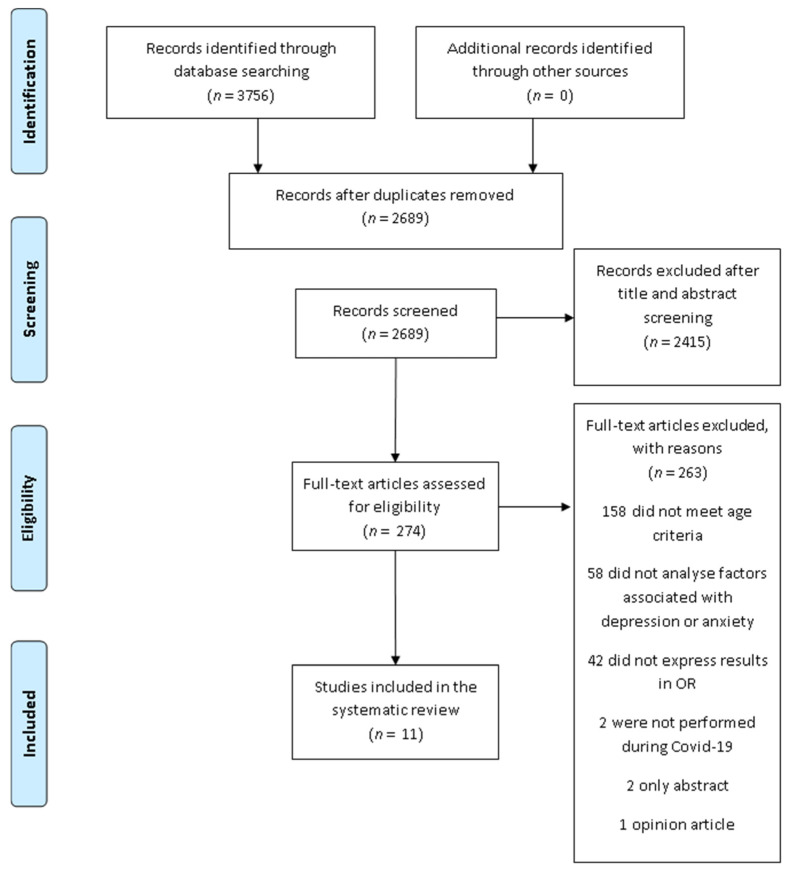
PRISMA flow-chart of the study selection process.

**Table 1 ijerph-18-11859-t001:** Studies included that analyse factors associated with depression.

Author and Year	Study Design	Time of the Study and Country	Participants Information	N	Age	Gender (%)	Depression Outcome Measure	Cutoff Score	Results (OR [95%CI])	Quality
Bobes-Bascarán et al. 2020 [21]	Cross-sectional	Between 19 and 26 March 2020, during the lockdown, in Spain.	Adults ≥ 60 recruited during the lockdown.	2194	Mean: 65.62 ± 5.05	54.6% Female	DASS-21 depression subscale	>4 points in the subscale	Multivariate Logistic Regression: female **2.004** (**1.559–2.575**), never married **0.665** (**0.456–0.970**); civil servant **0.530** (**0.293–0.957**); retired **0.539** (**0.311–0.934**); able to enjoy free time **0.268** (**0.148–0.488**); family/friends infected with COVID-19 **1.631** (**1.247–2.132**); past mental disorders **1.810** (**1.352–2.423**); current mental disorders **3.132** (**2.043–4.803**).	8
Carlos et al. 2020 [22]	Cross-sectional	Between 9 April 2020, one month after the imple- mentation of lockdown, and 4 May 2020, the day of transition to “phase 2”, in Italy.	Adults ≥ 65, stratified by level of neurocognitive deficit.	204	Median: 82	57.4% Female	GDS-5	≥2	Logistic regression model after controlling for other factors (age, dementia, new hobbies, digital literacy): sleep disturbances **2.29** (**1.06–4.93**); general health problems **2.45** (**1.16–5.16**); exercise **0.30** (**0.12–0.72**).	7
Robb et al. 2020 [23]	Cross-sectional	Between 30 April and 8 July 2020, in the United Kingdom.	Adults ≥ 50 from Cognitive Health in Ageing Register for Interventional and Observation Trials (CHARIOT).	7127	Mean: 70.6 ± 7.4	54.1% Female	Worsening or improving depression. Measured with the HADS-depression, with questions added to each ítem to self-report change from the beginning of COVID-19 restrictions.	≥4 answers for positive or negative change for considering depression worsening or improvement.	Multivariable model adjusted for age, sex, hypertension, hypercholesterolemia, type 2 diabetes, chronic obstructive pulmonary disease, cardiovascular disease and mental health conditions before lockdown: Worsened: women **2.46** (**2.10–2.89**); age **0.81** (**0.77–0.85**); single/widow/divorced **1.37** (**1.17–1.59**); smoker 1.41 (0.97–2.04); alcohol consumption (units p/w) **1.01** (**1.00–1.01**); poor sleep < once per week **2.00** (**1.51–2.65**); poor sleep 1–2 times per week **2.84** (**2.13–3.79**); poor sleep ≥3 times per week **6.91** (**5.21–9.15**); feeling lonely rarely **2.72** (**2.16–3.43**); feeling lonely sometimes **7.14** (**5.78–8.82**); feeling lonely often **17.24** (**13.20–22.50**); live alone **1.32** (**1.12–1.55**); friend/family social media contact 2–6 times per week 1.05 (0.90–1.23); friend/family social media contact, ≤once per week 0.99 (0.77–1.27). Improved: women 1.14 (0.77–1.69); age 0.89 (0.78–1.02); single/widow/divorced 0.65 (0.41–1.03); smoker 2.07 (0.94–4.57); alcohol consumption (units p/w) **0.97** (**0.95–0.99**); poor sleep < once per week 0.72 (0.44–1.18); poor sleep 1–2 times per week 0.78 (0.45–1.35); poor sleep ≥3 times per week 0.75 (0.40–1.42); feeling lonely rarely 0.62 (0.37–1.02); feeling lonely sometimes **0.49** (**0.26–0.91**); feeling lonely often 0.77 (0.30–1.99); live alone 0.62 (0.37–1.02); friend/family social media contact 2–6 times per week 0.72 (0.47–1.11); friend/family social media contact, ≤once per week 0.69 (0.35–1.36).	7
Di Santo et al. 2020 [24]	Cross-sectional	From 21 April to 7 May 2020, in Italy.	Adults ≥ 60 with mild cognitive impairment, part of a clinical trial.	128	Mean: 74.29 ± 6.51	81% Female	GDS-5	≥2	Multivariable logistic regression analysis: alone or poor relation with cohabitants **2.79** (**1.20–6.49**); poor sleep quality 1.85 (0.80–4.29); no pets 0.16 (0.02–1.20).	8
Do et al. 2020 [25]	Cross-sectional	Between 14 February and 2 March 2020, in Vietnam.	Adults aged 60–85.	928	Mean: 68.2 ± 6.51	56.3% Female	PHQ-9	≥10	Logistic regression model adjusted for age, marital status (in the group without COVID-19), education and social status: health literacy in subjects without COVID-19 1.02 (0.96–1.09); health literacy in subjects with COVID-19 **0.91** (**0.87–0.94**).	8
Li et al. 2021 [26]	Cross-sectional	Between 22 May and 15 July 2020, in China.	Adults ≥ 50 with psychiatric disorders.	1063	Mean: 62.8 ± 9.4	67.4% Female	PHQ-9	≥5 depression; ≥10 moderate to severe depression.	Binary logistic regression analysis: Depression: rural area 1.29 (0.88–1.89); having severe physical diseases 1.35 (0.85–2.15); poor treatment adherence 1.25 (0.88–1.78); difficulty attending psychiatric hospital 1.38 (0.95–1.99); schizophrenia 0.95 (0.51–1.77); organic mental disorder 0.57 (0.28–1.16); other psychiatric diseases **0.50** (**0.35–0.71**); education years 0.99 (0.94–1.03); insomnia (ISI score) **1.29** (**1.24–1.34**); pain score **1.14** (**1.03–1.25**). Combined depression and anxiety: rural area 1.36 (0.95–1.93); having severe physical diseases 1.49 (0.98–2.26); poor treatment adherence **1.42** (**1.03–1.95**); difficulty attending psychiatric hospital 1.37 (0.97–1.91); schizophrenia **0.50** (**0.26–0.97**); organic mental disorder 0.66 (0.34–1.29); other psychiatric diseases **0.53** (**0.38–0.73**); education years 0.98 (0.94–1.03); insomnia (ISI score) **1.19** (**1.16–1.23**); pain score **1.15** (**1.06–1.25**).	8
Kitani-Morii et al. 2021 [27]	Cross-sectional	From 22 April to 15 May 2020 during the state of emergency in Japan.	Adults with Parkinson’s disease and control group.	71 (39 Parkinson; 32 control)	Mean: 72.3 ± 10.9 (Parkinson); 66.4 ± 13.8 (control)	35% Female (Parkinson); 84% Female (Control)	PHQ-9	≥10	Multivariate logistic regression analyses in patients with Parkinson’s disease: male 5.66 (0.51–62.47); aged between 70 and 79 **0.61** (**0.05–0.08**); aged ≥80 0.19 (0.01–3.93); disease duration ≥5 years 1.01 (0.13–7.78); HY stage 3,4 10.17 (0.57–182.91); MDS-UPDRS part 2 **1.31** (**1.04–1.66**); L-dopa ≥600 mg 1.39 (0.13–15.26); dopamine agonist 9.33 (0.85–102.72).	8
McArthurt et al. 2021 [28]	Longitudinal Retrospective	Assessments from January 2017 to June 2020, in Canada.	Long-term care homes residents.	765	Mean: 81.4 ± 11.5	59.5% Female	DRS	≥3	Longitudinal Multivariate Model: age 1.00 (0.98–1.01); female **2.11** (**1.47–3.04**); lockdown 0.86 (0.66–1.11); being in the reference home “facility X” **0.45** (**0.27–0.74**); Alzheimer’s and other dementias **0.69** (**0.48–0.99**); CPS **1.55** (**1.18–2.04**); CPS 2 measure **0.92** (**0.88–0.96**); CHESS **1.17** (**1.07–1.29**); ABS **1.28** (**1.22–1.34**); ADL Hierarchy **1.11** (**1.00–1.24**).	10
Piskorz et al. 2021 [29]	Cross-sectional	From 15 June to 15 July 2020, in Mexico, Guatemala, El Salvador, Costa Rica, Cuba, the Dominican Republic, Venezuela, Colombia, Ecuador, Peru, Paraguay, Chile, and Argentina.	Adults with cardiometabolic disease were recruited during the lockdown.	4216	Mean: 60.35 ± 15.39	49.07% Female	DSM-5	1 positive answer to the main questions or 3 or more positive answers to the additional questions.	Multivariate logistic regression: female **1.72** (**1.40–2.11**); consuming ≥5 medications/day **1.29** (**1.00–1.66**); physical activity less than 100 minutes per week **1.36** (**1.10–1.67**); low fruits and vegetables consumption **1.46** (**1.05–2.03**); poor treatment adherence **1.43** (**1.10–1.85**); reduced food intake **2.10** (**1.68–2.62**).	7
Cigiloglu et al. 2021 [30]	Cross-sectional	40 days after the detection of the first national COVID-19 case and 30 days after curfew was declared in Turkey.	Adults ≥ 65 who had to remain at home during the pandemic.	104	Stratified by age group: 65–74, 72.1%; 75–84, 17.3%; ≥85, 10.6%.	41.3% Female	GDS-15	≥5	Multivariate logistic regression analysis: female 2.25 (0.89–5.64); age 1.53 (0.78–2.98); monthly income medium vs low 0.34 (0.08–1.46); monthly income high vs low **0.13** (**0.04–0.44**); number of chronic diseases 1.08 (0.71–1.65).	7
Bérard et al. 2021 [31]	Cross-sectional	From 17 April to 10 May 2020, with mean time (±standard deviation) in lockdown before interviews of 44 days (±6 days), in France.	Adults aged between 50–89 during lockdown were recruited from a previous population-based study (PSYCOV-CV).	536 (489 analysis of the factors associated with depression or anxiety)	Median: 67	52% Female	PHQ-9	>4	Multivariate logistic regression analysis: Depression or Anxiety: female gender **1.98** (**1.23–3.20**); home with balcony or terrace 0.21 (0.04–1.04); home with garden 0.29 (0.06–1.26); not in total agreement with the effectiveness of preventive measures **2.46** (**1.42–4.27**); feeling socially isolated during lockdown **1.68** (**1.05–2.67**); worsening relationship with a partner since the beginning of lockdown **5.24** (**2.11–13.0**); pre-lockdown diet quality score > median **0.51** (**0.31–0.85**); history of anxiety **7.34** (**4.45–12.1**).	3

Values in bold indicate statistically significant associations. GDS: Geriatric Depression Scale. PHQ: Patient Health Questionnaire. DSM: Diagnostic and Statistical Manual of Mental Disorders. DRS: Depression Rating Scale. ISI: Insomnia Severity Index. HY: Hoehn & Yahr. MDS-UPDRS: Movement Disorder Society Unified Parkinson’s Disease Rating Scale. CPS: cognitive performance scale. CHESS: Changes in End-Stage Disease, Signs and Symptoms. ABS: Aggressive Behavior Scale. ADL: activities of daily living.

**Table 2 ijerph-18-11859-t002:** Studies included that analyse factors associated with anxiety.

Author and Year	Study Design	Time of the Study and Country	Participants Information	N	Age	Gender (%)	Anxiety Outcome Measure	Cutoff Score	Results (OR [95%CI])	Quality
Bobes-Bascarán et al. 2020 [21]	Cross-sectional	Between 19 March and 26 March 2020, during the lockdown, in Spain.	Adults ≥ 60 recruited during the lockdown.	2194	Mean: 65.62 ± 5.05	54.6% Female	DASS-21 anxiety subscale	>4 points in the subscale	Multivariate logistic regression: age **0.876** (**0.800–0.960**); female **3.320** (**1.511–7.294**); able to enjoy free time **0.103** (**0.047–0.227**); more than 14 days with COVID-19 symptoms **7.584** (**1.398–41.146**); current mental disorders **6.202** (**3.005–12.799**).	8
Robb et al. 2020 [23]	Cross-sectional	Between 30 April and 8 July 2020, in the United Kingdom.	Adults ≥ 50 from Cognitive Health in Ageing Register for Interventional and Observation Trials (CHARIOT).	7127	Mean: 70.6 ± 7.4	54.1% Female	Worsening or improving anxiety. Measured with the HADS-anxiety, with questions added to each ítem in order to self-report change from the beginning of COVID-19 restrictions.	≥4 answers for positive or negative change for considering anxiety worsening or improvement	Multivariable model adjusted for age, sex, hypertension, hypercholesterolemia, type 2 diabetes, chronic obstructive pulmonary disease, cardiovascular disease and mental health conditions before lockdown: Worsened: women **2.42** (**2.06–2.85**), age **0.78** (**0.75–0.83**); single/widow/divorced **1.17** (**1.00–1.37**); smoker 1.16 (0.79–1.72); alcohol consumption (units p/w) 1.00 (1.00–1.01); poor sleep < once per week **1.81** (**1.34–2.45**); poor sleep 1–2 times per week **3.50** (**2.59–4.73**); poor sleep ≥3 times per week **7.67** (**5.69–10.33**); feeling lonely rarely **1.65** (**1.32–2.07**); feeling lonely sometimes **4.73** (**3.87–5.77**); feeling lonely often **10.85** (**8.39–14.03**); live alone **1.15** (**0.98–1.36**); friend/family social media contact 2–6 times per week **0.81** (**0.68–0.95**); friend/family social media contact, ≤ once per week 0.77 (0.59–1.00). Improved: women **1.7** (**1.36–2.16**); age 0.97 (0.89–1.04); single/widow/divorced **1.30** (**1.03–1.64**); smoker 1.36 (0.78–2.38); alcohol consumption (units p/w) 1.00 (0.99–1.01); poor sleep < once per week **0.53** (**0.41–0.69**); poor sleep 1–2 times per week **0.43** (**0.31–0.60**); poor sleep ≥3 times per week **0.41** (**0.28–0.61**); feeling lonely rarely **0.72** (**0.56–0.96**); feeling lonely sometimes **0.62** (**0.45–0.86**); feeling lonely often **0.53** (**0.28–0.99**); live alone 1.05 (0.82–1.35); friend/family social media contact 2–6 times per week **0.74** (**0.57–0.94**); friend/family social media contact, ≤ once per week 0.76 (0.51–1.11).	7
Di Santo et al. 2020 [24]	Cross-sectional	From 21 April to 7 May 2020, in Italy.	Adults ≥ 60 with mild cognitive impairment, part of a clinical trial.	128	Mean: 74.29 ± 6.51	81% Female	GAD-7	≥10	Multiple logistic models: subjective cognitive disorder **4.39** (**1.03–18.69**); cold/flu symptoms **4.01** (**1.13–14.24**); reduction in productive activities **4.42** (**1.10–17.76**); time spent searching information 2.45 (0.71–8.45).	8
Li et al. 2021 [26]	Cross-sectional	Between 22 May and 15 July 2020, in China.	Adults ≥ 50 with psychiatric disorders.	1063	Mean: 62.8 ± 9.4	67.4% Female	GAD-7	≥5 anxiety; ≥10 moderate to severe anxiety.	Binary logistic regression analysis: Anxiety: rural area 1.20 (0.86–1.68); having severe physical diseases **1.57** (**1.05–2.35**); poor treatment adherence **1.50** (**1.11–2.03**); difficulty attending psychiatric hospital 1.33 (0.96–1.84); schizophrenia 0.67 (0.37–1.22); organic mental disorder 0.78 (0.41–1.47); other psychiatric diseases 0.74 (0.54–1.01); education years 0.97 (0.93–1.01); insomnia (ISI score) **1.15** (**1.12–1.18**); pain score **1.11** (**1.02–1.20**). Combined depression and anxiety: rural area 1.36 (0.95–1.93); having severe physical diseases 1.49 (0.98–2.26); poor treatment adherence **1.42** (**1.03–1.95**); difficulty attending psychiatric hospital 1.37 (0.97–1.91); schizophrenia **0.50** (**0.26–0.97**); organic mental disorder 0.66 (0.34–1.29); other psychiatric diseases **0.53** (**0.38–0.73**); education years 0.98 (0.94–1.03); insomnia (ISI score) **1.19** (**1.16–1.23**); pain score **1.15** (**1.06–1.25**).	8
Kitani-Morii et al. 2021 [27]	Cross-sectional	From 22 April to 15 May 2020, during the state of emergency, in Japan.	Adults with Parkinson’s disease and control group.	71 (39 Parkinson; 32 control)	Mean: 72.3 ± 10.9 (Parkinson); 66.4 ± 13.8 (control)	35% Female (Parkinson); 84% Female (Control)	GAD-7	≥7	Multivariate logistic regression analysis in patients with Parkinson’s disease: male **17.12** (**1.13–257.27**); aged between 70 and 79 0.55 (0.04–7.69); aged ≥ 80 0.16 (0.01–3.08); disease duration ≥5 years 0.35 (0.04–3.08); HY stage 3,4 8.19 (0.52–128.74); MDS-UPDRS part 2 **1.36** (**1.07–1.72**); dopamine agonist 13.07 (0.81–210.16).	6
Cigiloglu et al. 2021 [30]	Cross-sectional	40 days after the detection of the first national COVID-19 case and 30 days after curfew was declared, in Turkey.	Adults ≥ 65 who had to remain at home during the pandemic.	104	Stratified by age group: 65–74, 72.1%; 75–84, 17.3%; ≥85, 10.6%.	41.3% Female	GAI	8/9	Multivariate logistic regression analysis: female **3.25** (**1.22–8.70**); age 0.89 (0.45–1.75); monthly income medium vs low 0.21 (0.04–1.19); monthly income high vs low **0.07** (**0.02–0.35**); number of chronic diseases 1.18 (0.75–1.86).	7
Bérard et al. 2021 [31]	Cross-sectional	From 17 April to 10 May 2020, with mean time (±standard deviation) in lockdown before interviews of 44 days (±6 days), in France.	Adults aged between 50 and 89 during lockdown were recruited from a previous population-based study (PSYCOV-CV).	536 (489 for the analysis of the factors associated with depression or anxiety)	Median: 67	52% Female	GAD-7	>4	Multivariate logistic regression analysis: Depression or Anxiety: female gender **1.98** (**1.23–3.20**); home with balcony or terrace 0.21 (0.04–1.04); home with garden 0.29 (0.06–1.26); not in total agreement with the effectiveness of preventive measures **2.46** (**1.42–4.27**); feeling socially isolated during lockdown **1.68** (**1.05–2.67**); worsening relationship with a partner since the beginning of lockdown **5.24** (**2.11–13.0**); pre-lockdown diet quality score > median **0.51** (**0.31–0.85**); history of anxiety **7.34** (**4.45–12.1**).	3

Values in bold indicate statistically significant associations. DASS: Depression Anxiety Stress Scale. GAD: General Anxiety Disorder. GAI: Geriatric Anxiety Inventory. ISI: Insomnia Severity Index. HY: Hoehn & Yahr. MDS-UPDRS: Movement Disorder Society Unified Parkinson’s Disease Rating Scale.

**Table 3 ijerph-18-11859-t003:** Joanna Briggs Institute tool for cross-sectional studies.

Cross-Sectional Studies	Inclusion Criteria	Participants and Setting	Exposition	Measurement of the Condition	Identify Confounding Factors	Deal with Confounding Factors	Outcomes	Statistical Analysis	Total
Author and Year									
Bobes-Bascarán et al. 2020s	+	+	+	+	+	+	+	+	8
Carlos et al. 2020	+	+	+	+	-	+	+	+	7
Robb et al. 2020	+	+	+	+	+	+	-	+	7
Di Santo et al. 2020	+	+	+	+	+	+	+	+	8
Do et al. 2020	+	+	+	+	+	+	+	+	8
Li et al. 2021	+	+	+	+	+	+	+	+	8
Kitani-Morii et al. 2021	+	+	+	+	-	-	+	+	6
Piskorz et al. 2021	+	+	+	+	+	?	+	+	7
Cigiloglu et al. 2021	+	?	+	+	+	+	+	+	7
Bérard et al. 2021	?	-	?	+	-	-	+	+	3

+ = Yes. - = No. ? = Unclear.

**Table 4 ijerph-18-11859-t004:** Joanna Briggs Institute tool for cohort studies.

Cohort Studies	Group Recruitment	Group Exposure	Exposure Measurement	Identify Confounding Factors	Deal with Confounding Factors	Not Exposure Previous the Study	Outcomes	Follow Up Time	Follow Up Complete	Strategies to Address Incomplete Follow up	Statistical Analysis	Total
Author and Year												
McArthur et al. 2021	+	+	+	+	+	-	+	+	+	-	+	10

+ = Yes. - = No.

## Data Availability

Not applicable.

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
