# Peer review of "Factors Associated with Depression and Anxiety in Adults ≥60 Years Old during the COVID-19 Pandemic: A Systematic Review"

_ijerph, 2021, doi:10.3390/ijerph182211859_

Round 1
Reviewer 1 Report
This is a well written manuscript that assesses the different factors related to depression and anxiety among the elderly during the first months of the COVID-19 pandemic. The authors have provided a comprehensive background in the introduction, the methods seem robust, the results are well discussed and the conclusion is evidence-driven.
Though, I believe the paper would benefit of some adjustments to make it more solid and sound:
- The study aimed to explore the factors associated with depression and anxiety during the COVID-19 pandemic. Though, it is unclear if (1) the authors hypothesise that the factors would be different from those identified previously to the pandemic and (2) if there were any factors that emerged (or seem more or less relevant) during the pandemic. I would suggest the authors clarify those points in the Introduction and in the Discussion sections, and, if relevant, in the Conclusion of the manuscript.
- This systematic review includes 11 articles from different countries (and regions). As the prevalence of depression and anxiety vary across countries and the containment measures had different stringency levels, probably different risk/protective factors would be found in different groups of countries. I would suggest the authors include information about the countries in Tables 1 and 2 and discuss it in the Discussion section.
- Similarly, the authors do not provide information about when the studies were conducted. The population reacted differently to the pandemic in different moments in time (in the first months, in face of the unknown, the population was scared and probably anxiety was high in most population groups, then as the pandemic evolved and the social and economic impacts became more obvious, probably other population groups felt an impact in their psychological well-being). As such, it would be important to know when each study was done, and it would be interesting to see if there were major differences in the factors for anxiety and depression in time.
- The design of the studies included in this systematic review not only limit authors to “detect changes in time of the psychological state or estimate the real impact that the pandemic and its containment measures may have had on depression and anxiety” (lines 265-267 Discussion section) but also limit understanding the causality between the factors and the outcomes: poor sleep quality may be a risk factor for anxiety, but the reverse may also be true, as anxiety may cause insomnia. As such the authors should carefully revise the manuscript in order to avoid stating possible/probable paths of causation and discuss that in the Limitations paragraph of the Discussion.
Minor comments:
- Introduction: Lines 54-55: “the prevalence of depression in older adults has increased, with values that range from 14.6% [22] to 49.7% [23]” – The authors should provide a reference for the statement related to the increase of the prevalence of depression, or otherwise reword it. Indeed, the references are from two cross-sectional studies that do not provide evidence for the trends of depression prevalence;
- Methods: The Boolean search strings should be provided in the Methods section (or in supplementary material);
- Results: Line 67: “The majority of the papers screened showed that the female gender” – The authors should provide the number. Lines 85-86: “association between being aged between 70 and 79 and depression in Parkin- 85 son’s disease patients (OR: 0.61 [0.05- 0.08])” – The reference category is unclear (having 70-79 years old is protective comparing to 60-69 or 80-89, or…?)
- There is a lapse in line 210: “weather” should be replaced by “whether”.
Author Response
Comments and Suggestions for Authors
This is a well written manuscript that assesses the different factors related to depression and anxiety among the elderly during the first months of the COVID-19 pandemic. The authors have provided a comprehensive background in the introduction, the methods seem robust, the results are well discussed and the conclusion is evidence-driven.
All the changes in the manuscript will be in Green.
Though, I believe the paper would benefit of some adjustments to make it more solid and sound:
- The study aimed to explore the factors associated with depression and anxiety during the COVID-19 pandemic. Though, it is unclear if (1) the authors hypothesise that the factors would be different from those identified previously to the pandemic and (2) if there were any factors that emerged (or seem more or less relevant) during the pandemic. I would suggest the authors clarify those points in the Introduction and in the Discussion sections, and, if relevant, in the Conclusion of the manuscript.
Thank you very much for your contributions to modify and expand the document, we hope we have solved them.
The following information has been added:
Lines 61-65: “The risk and protective factors of depression in older adults were described in a recent systematic review, which identified more homogeneously impairment, sleep disorders and chronic diseases as main risk factors and physical activity as a protective factor; however, this review was not focused on the time of Covid-19 pandemic, as the included studies were published between January 2000 and March 2020 [19].”.
Line 67: “during Covid-19”.
Lines 74-76: “We hypothesize that female gender, older age, poorer health conditions, loneliness and a lower socio-economic status may be factors associated with depression and anxiety during the pandemic.”
Line 180: “and on the contrary of our initial hypothesis”.
- If there were any factors that emerged (or seem more or less relevant) during the pandemic.
It has been modified by adding the following information:
Lines 59-61: “and reported that female gender, being aged <35, lower socio-economic status, higher risk of Covid-19 infection and longer media exposure were associated with psychological dis-tress.”.
- This systematic review includes 11 articles from different countries (and regions). As the prevalence of depression and anxiety vary across countries and the containment measures had different stringency levels, probably different risk/protective factors would be found in different groups of countries. I would suggest the authors include information about the countries in Tables 1 and 2 and discuss it in the Discussion section.
Thank you, the information about the countries has been added in Table 1 and 2.
- Similarly, the authors do not provide information about when the studies were conducted. The population reacted differently to the pandemic in different moments in time (in the first months, in face of the unknown, the population was scared and probably anxiety was high in most population groups, then as the pandemic evolved and the social and economic impacts became more obvious, probably other population groups felt an impact in their psychological well-being). As such, it would be important to know when each study was done, and it would be interesting to see if there were major differences in the factors for anxiety and depression in time.
the information about the countries has been Lines 268-271: “Also, it must be taken into account that the included studies were performed mostly within the first months since the beginning of the pandemic, thus considering the changes in public health policies and the evolution of Covid-19 pandemic, the results of this systematic review can only be extended to this period of time.”.
- The design of the studies included in this systematic review not only limit authors to “detect changes in time of the psychological state or estimate the real impact that the pandemic and its containment measures may have had on depression and anxiety” (lines 265-267 Discussion section) but also limit understanding the causality between the factors and the outcomes: poor sleep quality may be a risk factor for anxiety, but the reverse may also be true, as anxiety may cause insomnia. As such the authors should carefully revise the manuscript in order to avoid stating possible/probable paths of causation and discuss that in the Limitations paragraph of the Discussion.
The following information has been added:
Lines 275-276: “and similarly it also limits understanding the causality between the factors and outcomes studied.”.
Minor comments:
- Introduction: Lines 54-55: “the prevalence of depression in older adults has increased, with values that range from 14.6% [22] to 49.7% [23]” – The authors should provide a reference for the statement related to the increase of the prevalence of depression, or otherwise reword it. Indeed, the references are from two cross-sectional studies that do not provide evidence for the trends of depression prevalence.
It has been modified using:
Lines 53-55: “According to previous research, the prevalence of depression in older adults showed a significant increase from 7.2% to 19.8% since the beginning of the pandemic [17]”.
- Methods: The Boolean search strings should be provided in the Methods section (or in supplementary material).
The Boolean search strings have been provided in supplementary material.
- Results: Line 67: “The majority of the papers screened showed that the female gender” – The authors should provide the number. Lines 85-86: “association between being aged between 70 and 79 and depression in Parkin- 85 son’s disease patients (OR: 0.61 [0.05- 0.08])” – The reference category is unclear (having 70-79 years old is protective comparing to 60-69 or 80-89, or…?)
It has been modified using:
Line 69: “Five papers screened”.
Lines 86-89: “In addition, Kitani-Morii et al. [27] reported that being aged between 70 and 79, in comparison to having <70 years of age, represented a factor associated with depression in Parkinson’s disease patients (OR: 0.61 [0.05–0.08])”
- There is a lapse in line 210: “weather” should be replaced by “whether”.
It has been modified using:
Line 214: “whether”.
Thank you very much for the review and for giving us the opportunity to improve the article for publication. We hope we have provided a solution to all suggestions.

Reviewer 2 Report
Thank you very much for the opportunity to review the manuscript. This paper is very interesting. While I think this systematic review will us understand the mental health of them, I recommend its publication with minor revisions.
- In the pandemic, the wave of infection has occurred many times. The timing and method of lockdown varies different country, and the time of impact may vary. When is the study period for each of the 11 studies? You may add it to the table1 and table2 or describe it in the text “search strategy”
- Is there any reason to limit the factors associated with depression and anxiety to Odds Ratios only?
- Are the factors associated with depression and anxiety in older adults during pandemic different from those before the pandemic? It is easy for me understand if you add a more explanation, including previous research.
Author Response
Thank you very much for the review and for giving us the opportunity to improve the article for publication. We hope we have provided a solution to all suggestions.
All changes in the manuscript are underlined
Comments and Suggestions for Authors
Thank you very much for the opportunity to review the manuscript. This paper is very interesting. While I think this systematic review will us understand the mental health of them, I recommend its publication with minor revisions.
- In the pandemic, the wave of infection has occurred many times. The timing and method of lockdown varies different country, and the time of impact may vary. When is the study period for each of the 11 studies? You may add it to the table1 and table2 or describe it in the text “search strategy”
The information about when the studies were conducted has been added in Table 1 and 2.
Lines 268-271: “Also, it must be taken into account that the included studies were performed mostly within the first months since the beginning of the pandemic, thus considering the changes in public health policies and the evolution of Covid-19 pandemic, the results of this systematic review can only be extended to this period of time.”.
- Is there any reason to limit the factors associated with depression and anxiety to Odds Ratios only?
We thought it was easier to carry out this review including only the Odds ratio. Other types of association statistics could have been taken into account..
- Are the factors associated with depression and anxiety in older adults during pandemic different from those before the pandemic? It is easy for me understand if you add a more explanation, including previous research.
The following information has been added:
Lines 58-63: “The risk and protective factors of depression in older adults were described in a recent systematic review, which identified more homogeneously impairment, sleep disorders and chronic diseases as main risk factors and physical activity as a protective factor; however, this review was not focused on the time of Covid-19 pandemic, as the included studies were published between January 2000 and March 2020 [19].”.
